# Diagnosis of Laryngeal Pemphigus Vulgaris Can Be Facilitated Using Advanced Endoscopic Methods

**DOI:** 10.3390/medicina57070686

**Published:** 2021-07-06

**Authors:** Lucia Staníková, Martin Formánek, Pavel Hurník, Peter Kántor, Pavel Komínek, Karol Zeleník

**Affiliations:** 1Department of Otorhinolaryngology and Head and Neck Surgery, University Hospital Ostrava, 17. Listopadu 1790, 70800 Ostrava, Czech Republic; lucia.stanikova@fno.cz (L.S.); martin.formanek@fno.cz (M.F.); peter.kantor@fno.cz (P.K.); pavel.kominek@fno.cz (P.K.); 2Department of Craniofacial Surgery, Faculty of Medicine, University of Ostrava, Syllabova 19, 70300 Ostrava, Czech Republic; 3Department of Pathology, University Hospital Ostrava, 17. listopadu 1790, 70800 Ostrava, Czech Republic; pavel.hurnik@fno.cz

**Keywords:** pemphigus vulgaris, larynx, Narrow Band Imaging (NBI), IMAGE1-S, enhanced contact endoscopy (ECE)

## Abstract

*Background:* Isolated laryngeal pemphigus vulgaris (LPV) is rare; however, early diagnosis is crucial in determining its course and prognosis. This paper aims to describe mucosal vascular changes typical for LPV using advanced endoscopic methods, which include Narrow Band Imaging (NBI), IMAGE1-S video-endoscopy and enhanced contact endoscopy (ECE). *Materials and Methods:* Retrospective analysis of all laryngeal mucosal lesion examined using advanced endoscopic methods during 2018–2020 at tertiary hospital was performed. *Results:* Videolaryngoscopy examination records of 278 patients with laryngeal mucosal lesions were analyzed; three of them were diagnosed with LPV. Epithelial vascularization of LPV included specific pattern. Intraepithelial papillary capillary loops were symmetrically stratified and were organized into “contour-like lines”. This specific vascularization associated with LPV were different from other laryngeal mucosal pathologies. *Conclusions*: Using advanced endoscopic methods supports early diagnosis of LPV and accelerate the diagnosis and treatment.

## 1. Introduction

Pemphigus comprises a group of autoimmune blistering diseases with skin and mucosal manifestations. Pemphigus vulgaris (PV) is a mucocutaneous disease that presents as enanthemas and erosions, typically in the oral cavity. Other mucosal surfaces are less frequently involved [1]. Exfoliating blisters are painful, can bleed, and in severe cases, they can lead to rapid weight loss. Pemphigus may have an idiopathic origin, or in some cases may be a drug-induced reaction [2]. Rarely, particularly in patients with lymphoproliferative disorders (mostly non-Hodgkin lymphoma and chronic lymphocytic leukemia), PV is considered a paraneoplastic manifestation [1].

PV with primary laryngeal involvement and without other mucosal or skin lesions is extremely rare; to date, only a very limited number of case series and case reports have been published on this manifestation [3,4,5,6,7,8]. Laryngeal mucosal lesions, like leukoplakia and erythroplakia, typically arouse suspicion of a precancerous or cancerous lesion. Laryngeal pemphigus vulgaris (LPV) is not typically considered first in a differential diagnosis, although early diagnosis and treatment affect its course and prognosis.

It has been acknowledged that advanced endoscopy methods, like narrow band imaging (NBI), IMAGE1-S™ video-endoscopy and enhanced contact endoscopy (ECE), can facilitate the pre-histological diagnosis of mucosal lesions [9,10,11,12,13,14,15,16,17,18]. However, to date, no study has described a vascular pattern typical of LPV. Therefore, this short communication aimed to describe the vascular pattern typical of LPV and discuss differences from other laryngeal pathologies.

## 2. Materials and Methods

Retrospective analysis of all flexible videolaryngoscopy examination records performed with aim to evaluate laryngeal mucosal lesions during 2018 to 2020 period was performed. After flexible videolaryngoscopy, all patients underwent microlaryngoscopy with excision of mucosal lesion and its histopathology examinations. Patients with diagnosed LPV were selected and their videolaryngoscopy examination records (including advanced endoscopy methods) were compared with other findings to determine if vascular pattern of LPV differs or not.

## 3. Results

Videolaryngoscopy examination records of 278 patients with laryngeal mucosal lesions were analyzed; 3 of them were diagnosed with LPV (Figure 1).

Epithelial vascularization of mucosal lesions in patients with LPV included specific pattern. LPV manifested as flat lesions that rise slightly above the mucosal surface. The surface of LPV lesions was partly coated with different thicknesses of leukoplakia. In areas of little or no leukoplakia, intraepithelial papillary capillary loops (IPCLs), symmetrically stratified and organized into contour-like lines, were observed with advanced endoscopic imaging (NBI, IMAGE1-S). IPCLs were very thin and short, and they ran towards the surface. They appeared as small brown spots, symmetrically distributed, aligned in regular rows and without clear boundaries. The overall picture resembles a “contour line map” (Figure 2, Figure 3 and Figure 4). This appearance is very specific and was not observed in other laryngeal pathologies (laryngeal cancer, dysplastic changes, papillomatosis, polyps etc.) in our group of patients. Enhanced contact endoscopy (ECE) revealed these changes even more precisely (Figure 5).

## 4. Discussion

Laryngeal mucosal lesions have broad differential diagnosis; it extends from chronic laryngitis, on one end, to invasive cancer, on the other end. Because LPV is rare, typical signs that can distinguish LPV from other pathologies might be overlooked. However, modern advanced endoscopic methods enable more precise visualization of vascular changes and can be used in “prehistological diagnostics”. Our observations suggest that the mucosal changes visible using advanced endoscopic methods and described above as a “contour line map” are quite indicative of LPV.

Main differential diagnosis of LPV include highly dysplastic lesions, laryngeal cancer, and papillomatosis. Indeed, the signs of malignant transformation in epithelial capillaries include irregular IPCL arrangements, asymmetric capillary courses, and increasing numbers and diameters of branching vessels [12,13,19,20,21]. The papilloma vascularization is typically characterized by irregular shapes and calibers, and IPCLs that course with regular exophytic papillae with central vessels [11]. However, these features are not present in PV lesions.

LPV vascularization resembles the benign, more than the malignant type of vascularization. PV vascularization best corresponds to Type III of the new Ni classification system [18]. The only sign that might lead us to consider malignancy is the presence of perpendicular vessels. However, the type of vessels and the typical “contour line map” pattern should indicate a diagnosis of LPV.

Using of advanced endoscopic methods (NBI, IMAGE1-S, ECE) to assess mucosal lesions is analogy to using modern dermoscopy and capillaroscopy techniques to assess tumorous, inflammatory, and rheumatological skin pathologies [22,23].

From the clinical point of view, it is important to mention that LPV lesions localized in the larynx can cause deterioration of airway patency [24]. Therefore, LPV should be included in the differential diagnosis in case of upper respiratory obstruction.

This report has certain shortcomings. Because no one patient from our group was diagnosed with pemphigoid, we could not provide differential diagnosis of LPV and pemphigoid. Moreover, we could not identify any report about vascular patter of pemphigoid described using advanced endoscopic methods in the current literature. Moreover, future studies are needed to explore this topic. Because LPV is a rare disease, multi-institutional cooperation is necessary to obtain more robust data on the subject.

## 5. Conclusions

Advanced endoscopic methods can facilitate distinguishing LPV from other, more common entities that occur in the larynx, particularly cancerous lesions. LPV can be identified by a typical “contour line map” pattern. Suspicion of LPV can accelerate the diagnostic process and treatment.

## Figures and Tables

**Figure 1 medicina-57-00686-f001:**
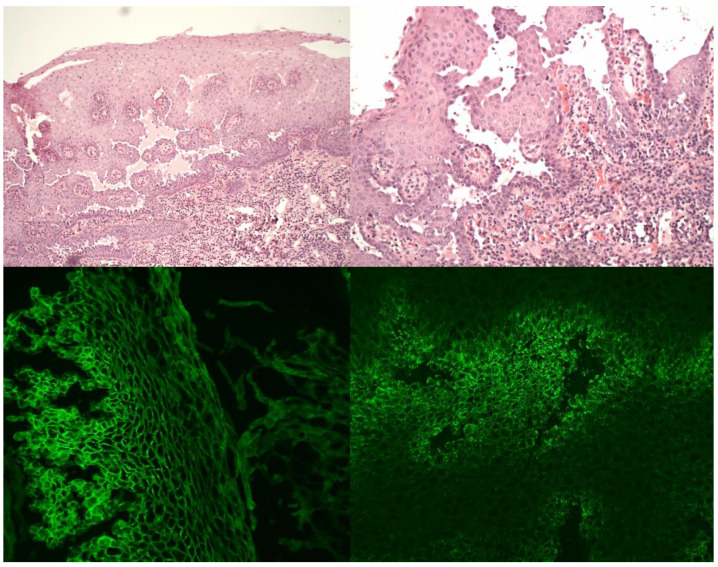
Histopathology and direct immunofluorescence of laryngeal pemphigus vulgaris specimens. Upper left—suprabasilar acantholytic blister with intraepithelial suprabasilar clefting; HE stain, 100× magnification. Upper right—suprabasilar acantholysis and plasma cell infiltrate among villous-like projections of lamina propria, covered by a few layers of epithelium; HE stain, 200× magnification. Lower left—intercellular linear deposition of IgG in direct immunofluorescence, DIF IgG, 200× magnification. Lower right—intercellular granular deposition of C3 in direct immunofluorescence, DIF C3, 200× magnification.

**Figure 2 medicina-57-00686-f002:**
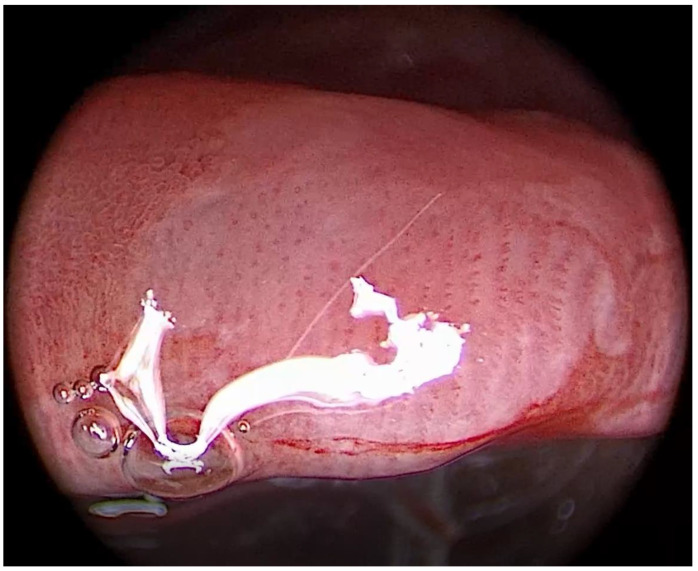
Patient 1, epiglottis. Typical endoscopic picture of laryngeal pemphigus vulgaris during laryngoscopy using IMAGE1-S, Clara + Chroma video-endoscopy. The overall picture resembles a “contour line map”.

**Figure 3 medicina-57-00686-f003:**
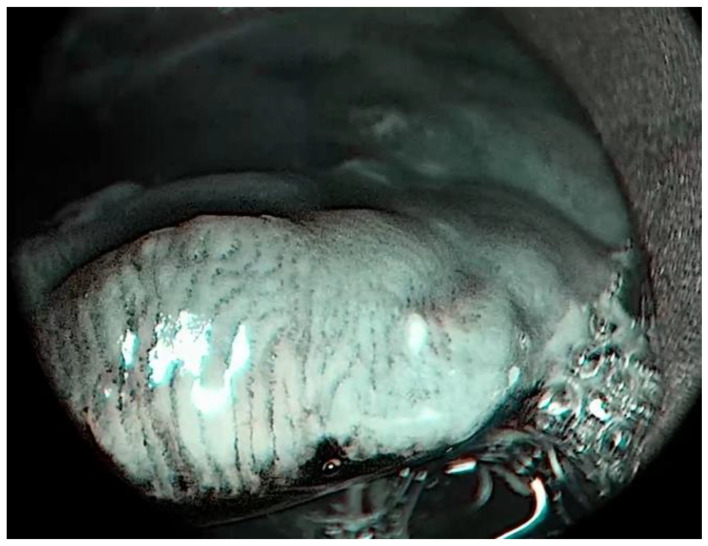
Patient 2, epiglottis. Epiglottis affected by laryngeal pemphigus vulgaris, imaged with IMAGE1-S, SPECTRA A mode. Intraepithelial papillary capillary loops (IPCLs) organized into contour lines are more accentuated with SPECTRA A modality.

**Figure 4 medicina-57-00686-f004:**
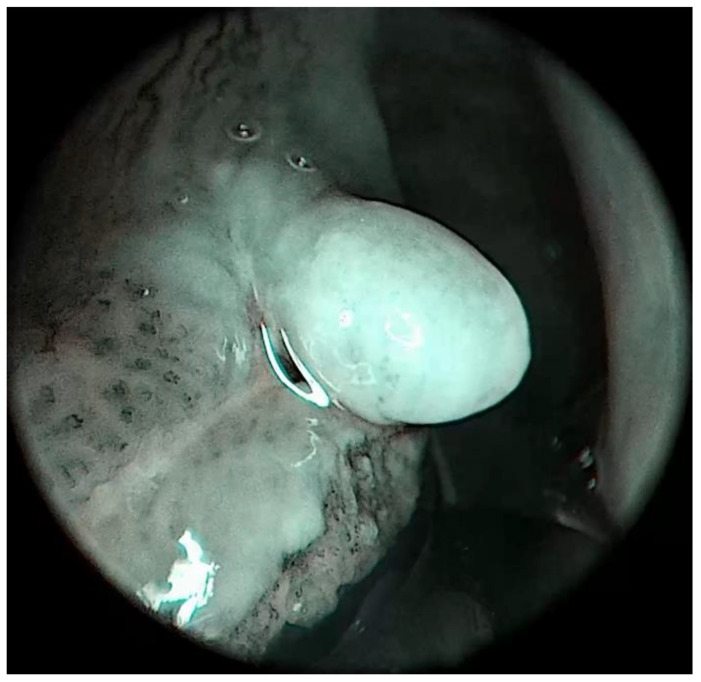
Patient 3, left vocal cord, posterior part. Laryngeal pemphigus vulgaris manifested as exophytic lesion covered with leukoplakia. Posteriorly to the exophytic lesion typical “contour line map” picture imaged with IMAGE1-S, SPECTRA A mode is present.

**Figure 5 medicina-57-00686-f005:**
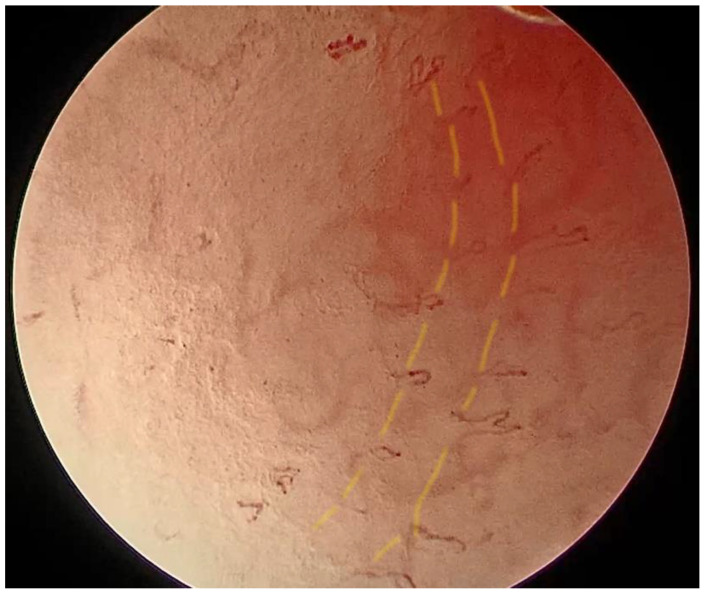
Epiglottis of patient 2 with laryngeal PV imaged with enhanced contact endoscopy (ECE), Clara + Chroma mode, 60× magnification. Very thin, short IPCLs are visible; they arise from the underlying vasculature and run toward the surface; they are symmetrically distributed and strictly stratified into regular rows (marked with yellow lines).

## Data Availability

The data presented in this study are available on request from the corresponding author.

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
