# Peer review of "Diagnosis of Laryngeal Pemphigus Vulgaris Can Be Facilitated Using Advanced Endoscopic Methods"

_medicina, 2021, doi:10.3390/medicina57070686_

Round 1

Reviewer 1 Report

The auuthors present the results of a single center retrospective analysis of all patients who underwent flexible videolaryngoscopy between 2018-2020 and had mucosal lesions, to identify vascularization patterns specific to pemphigus vuulgaris with lesions in the larynx.

The study is original and important, because differentials appearing with erosive laryngeal lesions particularly in the absence of cutaneous lesions of PV might be confusing. The authors identify leucoplakia, and intraepithelial papillary capillary loops thin and short, that run towards the surface, organized into contour like lines. These findings are of merit because no similar effort has been undertaken up to now, even though they refer to just 3 patients with LPV. However LPV is rare. 

The manuscript would improve if the authors mention in the discussion the importance of the early diagnosis of LPV that might be complicated with obstruction of the airways. ( Gregoriou et al JEADV 2015;29:1845-6)

In addition the authors could mention the analogy of capillaroscopy in the larynx with capillaroscopy and dermoscopy in inflammatory cuutaneuus lesions and the revolution this has started in diagnosis in dermatological diagnosis

Author Response

Dear Editors and Reviewers,

we are grateful for reviewer’s comments and reccomendations, which are very precious for us. Changes were made according to recommendations and are listed below step by step. Changes in re-submitted manuscript are in “tracking mode” in red colour.

Reviewer 1:

Query 1: The manuscript would improve if the authors mention in the discussion the importance of the early diagnosis of LPV that might be complicated with obstruction of the airways. (Gregoriou et al JEADV 2015;29:1845-6)

Changes were made according to recommendation – Page 5. Lines 127-129, reference 24 added and renumbered.

Query 2: In addition the authors could mention the analogy of capillaroscopy in the larynx with capillaroscopy and dermoscopy in inflammatory cuutaneuus lesions and the revolution this has started in diagnosis in dermatological diagnosis.

Changes were made according to recommendation – Page 5. Lines 124-126, reference 22 and 23 added and renumbered.

Reviewer 2 Report

A very intriguing case series about the use of advanced endoscopy to study the mucosal laryngeal pattern of this type of endoscopy.

I found this article very interesting, as it is very similar to what happened in dermatology some years ago, where dermoscopy was used also in the diagnosis of inflammatory condition, opening a subchapter of dermatology the inflammoscopy, that could be easily expanded also to ENT specialty using these new devices.

I have some queries:

Being the number of patients very limited. It would be useful (if available ) to have all endoscopic pictures of all patients, showing the common pattern in all of them.

Page 1 line 31 you should add: Pemphigus may have an idiopathic origin, or in some cases may be a drug-induced reaction" and cite an article such as: doi: 10.1111/dth.12748.

Thank You

Author Response

Dear Editors and Reviewers,

we are grateful for reviewer’s comments and reccomendations, which are very precious for us. Changes were made according to recommendations and are listed below step by step. Changes in re-submitted manuscript are in “tracking mode” in red colour.

Reviewer 2

Query 1: Being the number of patients very limited. It would be useful (if available) to have all endoscopic pictures of all patients, showing the common pattern in all of them.

Changes were made according to recommendation:

Picture 2 – Patient 1 (epiglottis)

Picture 3 – Patient 2 (epiglottis) – it was not mentioned in the original manuscript that it is different patient

Picture 4 – Patient 3 (left vocal cord)

Query 2: Page 1 line 31 you should add: Pemphigus may have an idiopathic origin, or in some cases may be a drug-induced reaction" and cite an article such as: doi: 10.1111/dth.12748.

Changes were made according to recommendation – Page 1. Lines 32-33, reference 2 added and renumbered all references

Round 2

Reviewer 2 Report

The authors responded to all queries. The paper is publishable